# Vript: A Video Is Worth Thousands of Words

**Dongjie Yang**[1], **Suyuan Huang**[2], **Chengqiang Lu**[3], **Xiaodong Han**[3],
**Haoxin Zhang**[3], **Yan Gao**[3], **Yao Hu**[3], **Hai Zhao**[1, *]
[1]Shanghai Jiao Tong University, [2] Beihang University, [3] Xiaohongshu Inc.
[1]`{djyang.tony@,zhaohai@cs.}sjtu.edu.cn`, [2]`huangsuyuan@buaa.edu.cn`,
[3]`{lusuo,shuweng,haoli9,yadun,xiahou}@xiaohongshu.com`

## Abstract

Advancements in multimodal learning, particularly in video understanding and generation, require high-quality video-text datasets for improved model performance. Vript addresses this issue with a meticulously annotated corpus of 12K high-resolution videos, offering detailed, dense, and script-like captions for over 420K clips. Each clip has a caption of ~145 words, which is over 10x longer than most video-text datasets. Unlike captions only documenting static content in previous datasets, we enhance video captioning to video scripting by documenting not just the content, but also the camera operations, which include the shot types (medium shot, close-up, etc) and camera movements (panning, tilting, etc). By utilizing the Vript, we explore three training paradigms of aligning more text with the video modality rather than clip-caption pairs. This results in Vriptor, a top-performing video captioning model among open-source models, comparable to GPT-4V in performance. Vriptor is also a powerful model capable of end-to-end generation of dense and detailed captions for long videos. Moreover, we introduce Vript-Hard, a benchmark consisting of three video understanding tasks that are more challenging than existing benchmarks: Vript-HAL is the first benchmark evaluating action and object hallucinations in video LLMs, Vript-RR combines reasoning with retrieval resolving question ambiguity in long-video QAs, and Vript-ERO is a new task to evaluate the temporal understanding of events in long videos rather than actions in short videos in previous works. All code, models, and datasets (Vript, Vript_CN, Vript_Multilingual) are available in `https://github.com/mutonix/Vript`.

## 1  Introduction

With the rapid development of multimodal learning [2, 3, 4], researchers are increasingly focusing on understanding [5, 6, 7] and generation [8, 9, 10] of the video modality. This has triggered a surge in demand for high-quality video-text datasets containing high-resolution videos and detailed captions. Compared to image-text pairs [11, 12], video-text pairs are harder to obtain and annotate. As a video has an additional temporal dimension, it contains more information than a single image. Additionally, a video often comprises numerous events, and each event can consist of several scenes. For instance, a travel vlog might feature events such as preparing for the journey and visiting various destinations. Each event can be depicted using different shots. Video captioning takes more labor for annotators to check through the whole video and write down thousands of words to annotate every

---

*  Corresponding author.

Submitted to the 38th Conference on Neural Information Processing Systems (NeurIPS 2024) Track on Datasets and Benchmarks. Do not distribute.

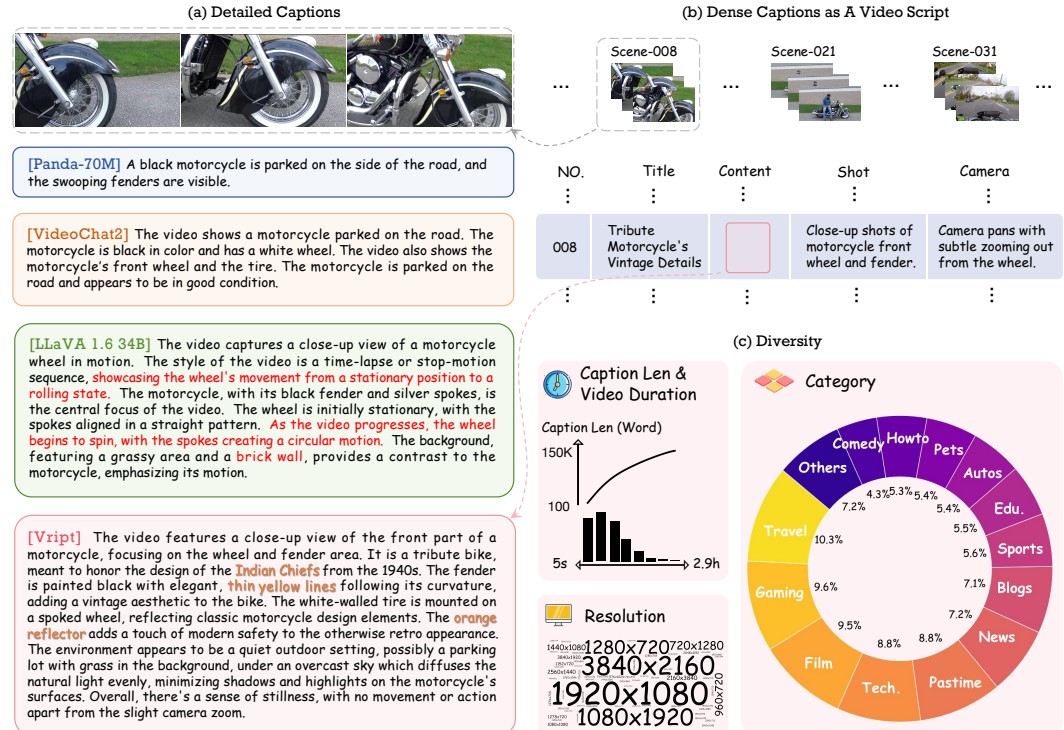

Figure 1: (a) We present a comparison between captions from our Vript and those produced by large multimodal models (LMMs). Compared to captions with hallucinations (marked in red) from LLaVA [1], Vript consists of the most detailed and accurate descriptions (marked in orange) for the videos. (b) Videos in Vript are densely annotated akin to video scripts, encompassing thousands of words. (c) Vript provides captions for open-domain videos in high resolution and various aspect ratios.

detail. Therefore, most previous video-text datasets only have short and coarse-grained descriptions for short video clips. For example, as shown in Table 1, WebVid-10M [13] and Panda-70M [14] comprise captions of 1~3 sentences for video clips shorter than 15 seconds.

To address the limitations of existing datasets, we construct a fine-grained video-text dataset called Vript, including 12K high-resolution videos (over 420K clips) annotated by GPT-4V [15]. The annotation process of Vript is inspired by the format of video scripts. A video script organizes the process of shooting a video consisting of multiple scenes. For each scene, we care not only about the content but also the camera operations, including shot types (medium shot, close-up, etc) and how the camera moves (panning, tilting, etc). Unlike most previous video-text datasets [13, 16], we densely annotate the untrimmed videos, and each scene in the video has a long caption of ~145 words. Besides the vision modality, we transcribe the voice-over into text and put it along with the video title to supplement background information, which greatly reduces the hallucinations in the captions.

Existing studies [17, 10, 1] report that detailed captions help improve better vision-language alignment. Most datasets [13, 14, 6] have short captions and are not densely annotated. Therefore, we can only align one short video clip with one short caption at a time during the training. To align more text with the video, we explore three paradigms that are not commonly used in vision-language alignment for videos: 1) Video-script alignment: We sample multiple successive scenes to form a longer video and concatenate the corresponding captions to create a "sub-script" as a longer text target. 2) Voice-over transcription: We combine the voice-over transcription and the video as input. 3) Video timestamp: We introduce the timestamps of both voice-over and video as additional information. Based on these, we train a video captioning model, dubbed Vriptor. Vriptor is good at generating dense captions both for short and long videos end to end and reaches SOTA performance in video captioning among open-source models.

Table 1: **Comparisons between Vript and other video-text datasets.** We divide the datasets into three parts. For the first part, the captions of these datasets come from subtitles (ASR) or descriptions scraped from the Internet. For the second part, the captions are collected by crowdworkers. For the third part, the captions are generated by multimodal models automatically.

| Dataset | Domain | Text Len | Clips | Duration | Resolution | Lang |
|---|---|---|---|---|---|---|
| HowTo100M [21] | Open | 4.0 | 136M | 134Kh | 240p | en |
| ACAV100M [22] | Open | - | 100M | 278h | - | en |
| HD-VILA-100M [23] | Open | 32.5 | 103M | 371Kh | 720p | en |
| WebVid-10M [13] | Open | ~12 | 10M | ~52Kh | 360p | en |
| YT-Temporal-180 [24] | Open | ~10 | 180M | - | 480p | en |
| MSVD [25] | Open | 8.7 | 1970 | 5.3h | - | en |
| MSR-VTT [16] | Open | 9.3 | 10K | 40h | 240p | en |
| DiDeMo [26] | Flickr | 8.0 | 27K | 87h | - | en |
| ActivityNet [27] | Action | 13.5 | 100K | 849h | 144p-720p | en |
| YouCook2 [28] | Cooking | 8.8 | 14K | 176h | - | en |
| VATEX [29] | Open | 15.2 | 41K | ~115h | - | en |
| HD-VG-130M [6] | Open | ~10 | 130M | ~180Kh | 720p | en |
| Panda-70M [14] | Open | 13.2 | 70M | 167Kh | 720p | en |
| InternVid [30] | Open | 17.6 | 234M | 760.3Kh | 720p | en |
| **Vript** | Open | ~145 | 420K | 1.3Kh | 720p-2K | en |
| **Vript-CN** | Open | ~150 | 293K | - | 720p-1080p | zh |
| **Vript-Multilingual** | Open | ~150 | 677K | - | 720p-1080p | multi |

Moreover, we propose Vript-Hard, a video understanding benchmark consisting of three tasks that are more challenging than most benchmarks [18, 16, 19]: 1) **Vript-HAL (Hallucination Evaluation)**: Vript-HAL is the first benchmark to comprehensively evaluate object and action hallucinations in video LLMs, providing the detailed ground truth 25x longer than MSR-VTT [16]. 2) **Vript-RR (Retrieval then Reasoning)**: Long video QA benchmarks[18, 19] ask questions about details in long videos that easily lead to ambiguity because the answers may vary in different timestamps. To solve this issue, we construct a long video reasoning task dubbed Vript-RR, by giving a hint for locating the relevant scene and then asking questions about the scene. Vript-RR features harder questions that need multi-hop reasoning and longer videos (2min~40min) than previous long video benchmarks, e.g., EgoSchema [18] (3min). 3) **Vript-ERO (Event Re-ordering)**: Different from previous benchmarks [7, 20] of temporal understanding that only care about chronological order of actions in short videos, we build a new challenging task called event re-ordering, requiring the model to sequence sampled events in long videos. In Vript-ERO, each video contains over 40 scenes on average and models need to re-order three of them in the correct order.

To sum up, we construct a high-quality video-text dataset called Vript, with dense and detailed captions for videos. Based on Vript, we train a top-performing video captioning model dubbed Vriptor. We propose Vript-Hard, a challenging video understanding benchmark that solves deficiencies in previous benchmarks, consisting of three tasks: Vript-HAL, Vript-RR, and Vript-ERO.

## 2 Related Work

**Video-text Dataset** Building powerful video foundation models [10, 31, 32, 1, 17] requires high-quality video-text datasets for vision-language alignment. In Table 1, we compare video-text datasets using different annotation methods. Datasets such as HD-VILA-100M [23] utilize subtitles as captions, which often can not precisely describe videos. Annotating videos manually [16, 27] gives accurate descriptions, yet it is challenging to scale up the dataset size. Recent datasets like HD-VG-130M [6] leverage large multimodal models (LMMs) to automatically generate captions but only short captions are provided due to the limitation of the model's ability. Compared to the above, Vript provides dense and detailed captions 10x longer for untrimmed videos by using GPT-4V [15].

**Video Understanding Benchmark**  Existing benchmarks [16, 25, 19, 18, 7] including captioning and QA tasks evaluate models on the short videos (<5min) and test the superficial understanding of the videos. In contrast, Vript-Hard scales up the videos to be much longer, e.g., Vript-RR (2min~40min) and Vript-ERO (2min~2h) and requires models to watch videos more carefully, e.g, Vript-HAL evaluating hallucinations of video LLMs and Vript-RR testing multi-hop reasoning ability.

# 3  Refine Video Captioning into Video Scripting

In the construction of Vript, our goal is to annotate a video as detailed as possible so that we can even visualize the video via the text description. For each scene in the video, we describe events with detailed actions and interactions rather than coarse-grained descriptions. Besides events, we record more details: the appearance of all objects and characters, environment, light, video style, etc.

In addition to the static description above, we inspect how the camera moves and shoots the scenes (Camera language).  Previous works [14, 6, 13] leverage the pipeline of describing an image to describe a video, ignoring the cameras. For a video clip about a man riding a bike, if we only describe what is in the frames, we can say "A man in a dark blue shirt is riding a black bike along the road". However, to be specific, we actually observe "As the camera pans to a close-up shot, a man in a dark blue shirt is riding a black bike. As the camera zooms out, we can see an overview of a man riding along the road with mountains behind him." Thus, to enhance the description of a video, it is necessary to record the camera language in addition to the content.

Combining both static description and camera language is like how we write a scene in a video script. In Vript, following the format of the video script, we first split the video into scenes using the PySceneDetect [2] and annotate each scene with static description and camera language, dubbed Video Scripting. We select 10K YouTube long videos from HD-VILA-100M [23] and collect 1.5K short videos from YouTube Shorts and TikTok from the Internet. We leverage the advanced multimodal model, GPT-4V [15], to annotate the following items for each scene: 1) title: a brief summarization of the scene within 10 words; 2) content: detailed description of around 150 words; 3) shot type: full view, close-up, etc; 4) camera movement: panning, zooming, etc. To make a "full" script of a video, we densely annotate the untrimmed videos (lasting from 5s to 2.9h) from the start to the end.

Besides video frames, we also add more external information to assist the annotation. We leverage the voice-over transcribed by the Whisper model [33] and also the video title, which helps the model to know what the original video is about. This external information greatly reduces the hallucinations and improves the caption granularity, helping the models to better understand what is happening in the video rather than what they have seen visually. For example, as shown in Figure 2, by watching the frames of Scene-010, we can not infer what ingredients are added to the bowl with the spoon and the squeeze bottle. The highlighted words from the voice-over illustrate they are mayonnaise and mustard, which improves the granularity of the caption shown in the top-right panel.

# 4  Vriptor: A Long Video Is Worth Thousands of Words

In the common paradigm of vision-language alignment for video foundation model training, assuming the batch size is 1, we align one video with one text caption.  Existing video-text datasets like Panda-70M [14] and WebVid-10M [13] only have brief captions where inadequate details result in suboptimal vision-language alignment. To alleviate this issue, we showcase how we can align more text with videos by training on the Vript dataset. We explore three not commonly used paradigms beyond the common one. Based on these, we train the Vriptor, a powerful model for video captioning, which reaches SOTA performance among open-source video LLMs.

---

[2]https://github.com/Breakthrough/PySceneDetect

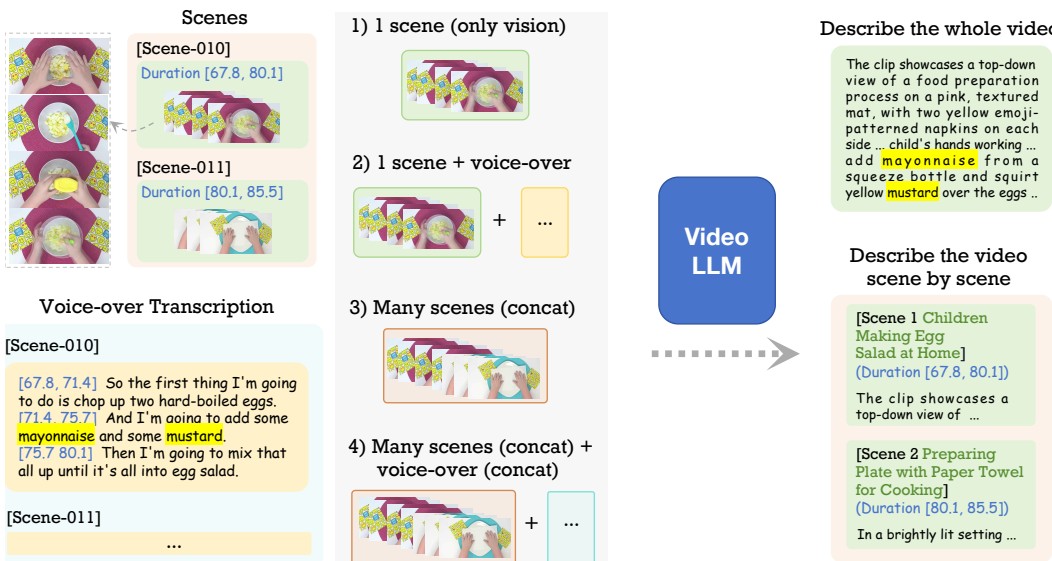

Figure 2: The input and output combinations of Vriptor training.

## 4.1 Method

**Video-Script Alignment**   If videos are densely annotated, a possible way to increase the amount of text for alignment is to concatenate captions of multiple successive clips. Though clips can be easily concatenated to create a longer video, captions are annotated separately so that the concatenated caption may not have coherence in the semantics. Inspired by video scripts, we reformulate the successive captions into scenes of the video script. In the right panel of Figure 2, a script in Vript with multiple scenes is coherent in the semantics despite they are annotated separately because: 1) each scene caption is very detailed and has similar descriptions for the shared background or context and 2) title of each scene acts as a separator rather than concatenating them directly. In Vript, We can easily sample several successive clips to create a "sub-script", e.g., 10 successive clips with corresponding "sub-script" containing about 1.5K words, which is nearly 100x longer than short captions.

**Voice-over Transcription**   We add voice-over transcription as the additional speech modality. As the Vript is annotated with joint input of voice-overs and video frames, the captions contain information that comes from the voice-over as shown in Figure 2.

**Video Timestamp**   Commonly video LLMs [7, 34] implement a certain sampling strategy to extract multiple frames as the video input. These models are weak in time awareness as they only know the order of frames but do not know how long the frames last. We find that timestamps are crucial for the video-script alignment of multiple scenes. As shown in Figure 2, we add two kinds of timestamps in the text format: voice-over timestamps in the input and video timestamps in the output caption. Predicting the timestamps of the video helps the model to know the start and the end of each scene.

## 4.2 Experiment and Analysis

We aggregate these paradigms to train Vriptor. In Figure 2, we combine four types of inputs and outputs: 1) 1 scene → 1 caption; 2) 1 scene + voice-over → 1 caption; 3) many scenes → 1 script; 4) many scenes + voice-over → 1 script. We add the timestamp information for all four types. We train the Vriptor based on ST-LLM [35] for two stages. We evaluate the captioning ability of the Vriptor on the Vript-HAL and the MSR-VTT [16], where the Vript-HAL and metrics are introduced in Sec 5.1 later. More details of training Vriptor can be checked in Appendix D.

**Video-Script Alignment Helps Model Watch More**  As shown in Figure 2, Vriptor supports two types of instructions: describe the whole video and scene by scene. For the whole-video instruction, Vriptor gives a general description of 100~150 words. For the scene-by-scene instruction, Vriptor gives a dense description of the video with each scene of 100~150 words. In Table 2, compared to the whole-video description, Vriptor gives more details of the video in the scene-by-scene description with an increasing recall in the Vript-HAL and the MSR-VTT as the number of output scenes increases. However, as the captions get longer and more detailed (more scenes), models are easier to generate hallucinations with a drop in precision. In Figure 3, we showcase the ability of Vriptor to caption long videos with longer texts. Models like VideoChat2 [7] only give a relatively fixed length of captions for videos of different lengths. Vriptor-S (scene-by-scene) can scale up the caption length as the video gets longer, just like writing a longer video script.

Table 2: Different strategies of video-script alignment and voice-over transcription.

| Strategy | Vript-HAL | | | MSR-VTT |
|---|---|---|---|---|
| | Precision | Recall | F1 | Recall |
| 2 scenes | **75.8** | 40.9 | 53.1 | 122.0 |
| 3 scenes | 74.1 | 49.5 | 59.4 | 135.8 |
| 4 scenes | 72.3 | 55.8 | 63.0 | 138.1 |
| 5 scenes | 71.4 | **57.5** | **63.7** | **139.5** |
| Whole | 79.1 | 26.8 | 40.0 | 83.0 |
| Whole (voice) | **80.3** | **27.7** | **41.1** | - |

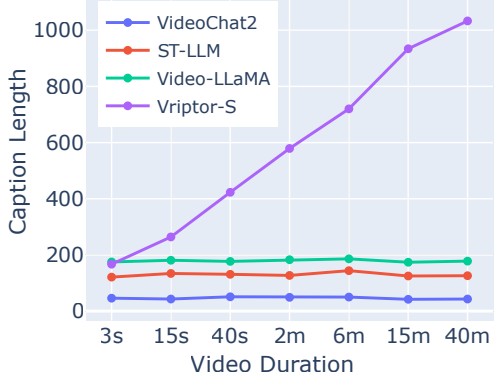

Figure 3: Caption lengths for videos of different durations.

**Voice-overs Help Model Understand What They Are**  In the last two rows in Table 2, we showcase the increments in both precision and recall that the model can give more detailed and accurate descriptions with the help of voice-over. We also observe a 14% increment in the proportion of proper nouns of all nouns in the captions. This suggests that the model is capable of inferring the names of objects rather than only their appearance by analyzing the voice-over.

**Timestamps Help Model Know the Starts and the Ends**  To verify the effectiveness of adding timestamps, we also train another model without adding timestamps. Comparing these two models, we find the improvement is minor in whole-video description but significant in scene-by-scene description. The model with timestamps is less likely to generate duplicated descriptions from previous scenes because it can understand the start and end of each scene and identify which scene corresponds to which period. Besides, the model with timestamps gives more detailed captions with a 12% higher recall on Vript-HAL while the model without timestamps is more likely to forget to describe some parts of the videos.

## 5   Vript-Hard

As multimodal models advance in performance, a more challenging benchmark is required to evaluate their capabilities. We propose a hard video understanding benchmark, dubbed Vript-Hard, consisting of three challenging tasks: HAL (Hallucination Evaluation), RR (Retrieval then Reasoning), and ERO (Event Re-ordering). We evaluate a large range of image LLMs, namely BLIP2 [36], InstructBLIP [37], Qwen-VL [38], LLaVA 1.6 34B [1], and video LLMs, namely VideoChatGPT [20], Video-LLaMA [32], VideoChat [31], VideoChat2 [7], ST-LLM [35], PLLaVA 7B [39], VILA-1.5 8B [40]. For open-source image and video LLMs, we sample 4 and 16 frames (following VideoChat2) per video respectively. We also evaluate sophisticated close-source models, namely Claude 3-Sonnet and Opus [41], GPT-4V [15], Claude 3.5-Sonnet [41], Gemini-1.5-Pro [42], GPT-4O [43]. For the close-source models, we sample 10 frames per video because GPT-4V can only accept a maximum of

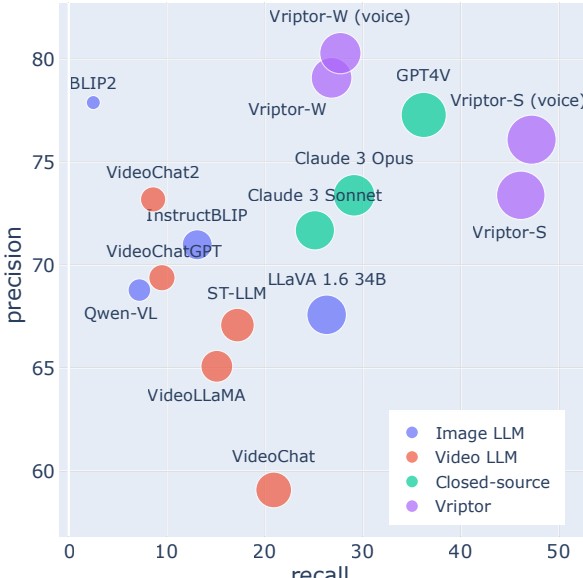

Figure 4: The precision and recall scores of various models on Vript-HAL. The sizes of the circles stand for the F1 values. The full results can be checked in Table 6.

10 frames. We also introduce a special setting for GPT-4O that we sample with 1 fps or a maximum of 100 frames (1 fps/100 fr). More details about Vript-Hard can be checked in Appendix E.

## 5.1    Vript-HAL: A Hallucination Evaluation Benchmark for Video LLMs

**Evaluating Hallucinations in Video LLMs**    Previous researchers [44, 45, 46] have explored methods to detect and evaluate hallucinations of powerful image LLMs. Similar to image LLMs, current video LLMs have a deeper understanding of videos and a stronger ability to generate more detailed captions for videos but also suffer from severe hallucinations. If we ask the video LLMs to describe a video, they may misread the objects and actions and generate a description with hallucinations. Captioning benchmarks, e.g., MSR-VTT [16] and MSVD [25], consist of short captions of no more than 10 words, giving superficial video descriptions without details. Thus we can not use them to evaluate hallucinations if many objects and actions are not included in the ground truth. To fill this gap, we construct Vript-HAL, a benchmark to evaluate object and action hallucinations in the video captions. Each video in Vript-HAL is annotated with two captions separately, approximately 250 words each, which are 25x longer than those in MSR-VTT. By building such strong ground truth captions, we can check if the video LLMs generate hallucinations in the captions.

**Hallucination Evaluation Metrics**    Traditional metrics, such as BLEU [47], ROUGE [48], and CIDEr [49], focus on word-for-word precision by measuring the token similarity between the predicted and ground truth texts, which are not suitable for evaluating if the objects and actions are correctly described. Following previous works [50, 51], we evaluate whether the nouns (objects) and verbs (actions) are correctly described in the captions by using the precision score. In addition to evaluating accuracy through precision, it is noted that various models give descriptions varying in length and detail. We observe that shorter captions typically include fewer details thus tending to contain fewer hallucinations. To balance this, we introduce the recall score, which measures how many objects and actions in the ground truth are correctly described. We calculate the F1 score as the comprehensive score of hallucination evaluation as follows:

$$\mathcal{P}(\mathbf{p}, \mathbf{g}) = \frac{\#\{\mathbf{p} \cap \mathbf{g}\}}{\#\{\mathbf{p}\}}, \quad \mathcal{R}(\mathbf{p}, \mathbf{g}) = \frac{\#\{\mathbf{p} \cap \mathbf{g}\}}{\#\{\mathbf{g}\}}, \quad F_1 = 2 \cdot \frac{\mathcal{P} \cdot \mathcal{R}}{\mathcal{P} + \mathcal{R}}, \tag{1}$$

where $\#\{\mathbf{p}\}$ and $\#\{\mathbf{g}\}$ represent the number of objects and actions described in the prediction and ground truth caption respectively. We leverage the SpaCy [3] to extract the nouns, proper nouns, and verbs as the objects and actions. $\#\{\mathbf{p} \cap \mathbf{g}\}$ represents the number of objects and actions that are correctly described in the prediction. We then encode the objects and actions into word embeddings using the sentence-transformers [4]. Instead of using the exact match, for each object or action, we consider it to be correctly described if the cosine similarity between the prediction and the ground truth is greater than 0.5. It is noted that using similarity may result in many-to-one matching because objects or actions with similar meanings in the prediction are all matched by one object or action in the ground truth, potentially yielding a score greater than 1 if the prediction is much longer than the ground truth, e.g., the recall score in MSR-VTT in Table 2.

**Evaluation**    We evaluate a large range of models on Vript-HAL, including image LLMs supporting multiple image inputs and video LLMs. From Figure 4, we observe some models, e.g., BLIP2 and VideoChat 2 have fewer hallucinations only because they give shorter captions containing fewer details. Vriptor-W (whole-video) giving general descriptions has a higher precision while Vript-S (scene-by-scene) giving dense descriptions describes more details in the videos with a higher recall. Both models have performance on par with the GPT-4V in video captioning.

## 5.2   Vript-RR: A Hard Reasoning Benchmark for Long Video Understanding

**Retrieving the Scene then Reasoning the Answer**    If we ask about details in the long video, we may encounter ambiguity in the questions that: 1) there are multiple answers that match the question in the different timestamps; 2) the answer changes as time goes on. The ambiguity issue can be commonly seen in the long video understanding benchmarks, e.g., EgoShecma [18]. We propose Vript-RR (Retrieval then Reasoning), a long video reasoning benchmark that has no such worries. Different from these benchmarks [19, 7, 18] that only provide questions, we first give a hint for the model to locate the scene in the video that the question refers to. The hint is a detailed description of the relevant scene. We then ask the question based on the scene, which eliminates the ambiguity. In practice, as shown in Figure 7, we input the hint and the question along with the entire video **together**, and the models directly output the answer, which is an end-to-end process. We carefully craft the hints to ensure the model can not find short paths through hints. We design various questions for Vript-RR to evaluate the different capabilities of video LLMs, where each question requires at least one reasoning step or additional processing, e.g., text reading, and meticulous inspection of details.

**Evaluation**    Vript-RR consists of two subtasks differing in the video inputs: one is inputting the whole videos and another is directly inputting the related scenes. Vript-RR provides questions both in multiple-choice and open-ended formats. For the open-ended outputs, we leverage the GPT-4 turbo [15] as the judge [52] to evaluate if the answer is correct by comparing the prediction with the ground truth. As shown in Table 3, the "Scene" columns represent using the related scene as input, which is an easier task because the models do not need to retrieve across the entire video to find the related scene. The results of the "Scene" columns mainly showcase the models' video reasoning ability. For "Whole" columns using the whole video as input, we require models to first find the relevant scenes using the hint, requiring the additional long video understanding ability. The closed-source models like GPT-4V and Claude 3 have better performance than open-source video LLMs.

**Finding A "Needle" In A "Timestack"**    For each video in Vript-RR, we design the questions for scenes extracted from four various timestamps, corresponding to 15%, 40%, 60%, and 85% of the video respectively. We want to explore whether the temporal positions of scenes in the long video will influence the results of Vript-RR. We describe it as finding a "needle" in the "timestack", whose name is derived from the "needle-in-a-haystack" task [53] for testing the long-context ability of LLMs. We require models to go through visual tokens instead of text tokens to find the "needles" (related scenes). In the "needle-in-a-haystack" task, there is a phenomenon that the model performance drops

---

[3] https://spacy.io/. We use the largest model *en_core_web_lg*.

[4] https://www.sbert.net. We use the top-performing embedding model *all-mpnet-base-v2*.

significantly when the "needle" falls between 15% and 85% of the long context, particularly when the text length exceeds at least 16K tokens. As shown in Figure 5 (a), though the number of visual tokens is significantly smaller than 16K, performance drops are also observed for most of the models if the scenes fall in the middle of the visual tokens (40% and 60% of the video).

Table 3: The metric of Vript-RR and Vript-ERO is accuracy. In Vript-RR, "M" and "O" stand for multiple-choice and open-ended questions respectively. In Vript-ERO, "@x" denotes the number of positions correctly predicted in the order of three shuffled scenes at different timestamps.

| Model | Vript-RR | | | | Vript-ERO | | |
|---|---|---|---|---|---|---|---|
| | Scene-M | Scene-O | Whole-M | Whole-O | @1 | @2 | @3 |
| VideoChatGPT [20] | 34.2 | 28.9 | 29.6 | 17.8 | - | - | - |
| Video-LLaMA [32] | 38.2 | 19.7 | 28.3 | 14.5 | - | - | - |
| VideoChat [31] | 33.6 | 23.0 | 22.4 | 15.1 | 46.2 | 17.1 | 17.1 |
| VideoChat2 [7] | 52.0 | 32.2 | 42.1 | 22.4 | - | - | - |
| ST-LLM [35] | 43.4 | 34.9 | 33.6 | 26.3 | - | - | - |
| PLLaVA 7B [39] | 62.5 | 46.1 | 55.3 | **36.2** | - | - | - |
| VILA-1.5 8B [40] | **75.0** | **48.7** | **55.3** | 32.3 | - | - | - |
| Claude 3-Sonnet [41] | 60.5 | 53.9 | 56.6 | 42.1 | 67.9 | 24.6 | 19.4 |
| Claude 3-Opus [41] | 63.8 | 60.52 | 60.5 | 43.4 | **70.2** | 26.9 | 23.9 |
| GPT-4V [15] | **80.9** | **75.0** | **71.7** | **61.0** | 59.2 | **28.4** | **27.7** |
| Claude 3.5-Sonnet [41] | 80.9 | 59.2 | 54.6 | 42.8 | 56.7 | 18.7 | 6.7 |
| Gemini-1.5-Pro [42] | 85.1 | 68.5 | 59.9 | 47.5 | 35.1 | 18.2 | 9.1 |
| GPT-4O [43] | 92.1 | 77.5 | 72.4 | 54.6 | 75.0 | 32.6 | 32.6 |
| GPT-4O (1 fps/100 fr) | 91.4 | 78.2 | 78.5 | 66.0 | 81.0 | 40.2 | 38.6 |

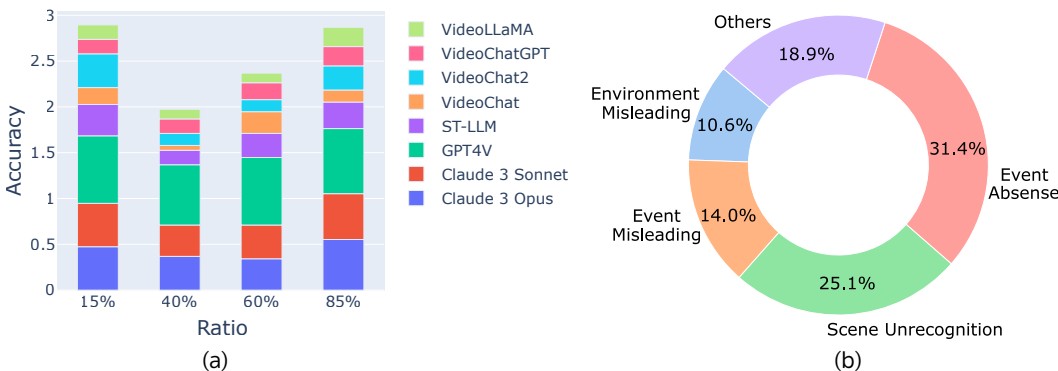

Figure 5: (a) The accuracies of Vript-RR questions regarding scenes at different timestamps (15%, 40%, 60%, and 85% of the video). (b) The reasons why the models (GPT-4V, Claude 3-Sonnet, and Opus) sequence the events inaccurately in Vript-ERO.

## 5.3  Vript-ERO: A Temporal Understanding Benchmark of Long Videos

**Re-ordering the Events in Different Timestamps**    There have been some benchmarks [5, 19, 7] that test the temporal understanding ability of the models. Unfortunately, they focus on asking questions about the temporal order of the actions happening in a short clip but few explore the temporal understanding of events in the long videos. To fill the gap, we propose the Vript-ERO (Event Re-ordering) task. We sample three distinct scenes (lasting 10s on average) in different timestamps from a long video (varying from 2min to 2h) and shuffle their chronological order. Given the long

video and the detailed descriptions of shuffled three scenes, the model is required to give the correct temporal order of the scenes based on the understanding of the entire video.

**Evaluation**    In Table 3, "-" means these models fail to give answers. Different from previous tasks that only have questions, Vript-ERO also contains long descriptions of scenes, which indicates these models are weak in processing long instructions. For models having scores, they only give the correct orders of all three scenes (@3) in about 20% of questions. In Figure 5 (b), we collect answers to the questions that are answered incorrectly and analyze the reasons. We observe that the models can be easily misled by the provided descriptions. For example, environment descriptions like sunlight may imply the morning or evening, however, these events may come from different days in the video rather than sequentially happening in one day. In 31.4% of cases, some events are absent in the input frames due to the limitation of the number of input images for models like GPT-4V. Besides, in 25.1% of cases, the models do not recognize which scene to be sequenced based on the descriptions. For the GPT-4O (1 fps/100 fr), which operates at 1 fps or 100 frames, an increased number of frames within the input significantly enhances the overall scores. This is because the probability of the relevant events being omitted decreases with a larger input frame count.

# 6    Conclusion

We introduce Vript, a high-quality video-text dataset consisting of dense and detailed captions for videos. Based on Vript, we train Vriptor, a top-performing video captioning model among open-source models. Besides, we propose Vript-Hard, a challenging video understanding benchmark evaluating hallucinations and the long video understanding ability of video LLMs.

# Acknowledgement

Dongjie Yang and Hai Zhao are with the Department of Computer Science and Engineering, Shanghai Jiao Tong University; Key Laboratory of Shanghai Education Commission for Intelligent Interaction and Cognitive Engineering, Shanghai Jiao Tong University; Shanghai Key Laboratory of Trusted Data Circulation and Governance in Web3.

This paper was completed during Dongjie Yang's internship at Xiaohongshu Inc. and was partially supported by the Joint Research Project of the Yangtze River Delta Science and Technology Innovation Community (No. 2022CSJGG1400).

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

# A Limitation and Potential Risk

## A.1 Limitation

We utilize advanced models like GPT-4V [15] to annotate the data in this paper, where GPT-4V sometimes generates inaccurate descriptions and hallucinations. For the Vript dataset, we do not check whether the descriptions are correct or not manually, where there may exist hallucinations from GPT-4V. For Vript-Hard for evaluation, we have carefully inspected and revised the content that is inaccurate and inappropriate manually, reducing the errors to the greatest extent.

## A.2 Potential Risk

For the Vript and Vript-Hard, we collect videos from YouTube and TikTok that may contain personal information and copyrighted items. Therefore, people using the Vript or Vript-Hard should respect the privacy and copyrights of the video owner and strictly agree to the license in Appendix B.

# B License

By downloading or using the data or models, you understand, acknowledge, and agree to all the terms in the following agreement.

**ACADEMIC USE ONLY**   Any content from Vript/Vript-Hard dataset and Vriptor model is available for academic research purposes only. You agree not to reproduce, duplicate, copy, trade, or exploit for any commercial purposes

**NO DISTRIBUTION**   Respect the privacy of personal information of the original source. Without the permission of the copyright owner, you are not allowed to perform any form of broadcasting, modification or any other similar behavior to the data set content.

**RESTRICTION AND LIMITATION OF LIABILITY**   In no event shall we be liable for any other damages whatsoever arising out of the use of, or inability to use this dataset and its associated software, even if we have been advised of the possibility of such damages.

**DISCLAIMER**   You are solely responsible for legal liability arising from your improper use of the dataset content. We reserve the right to terminate your access to the dataset at any time. You should delete the Vript/Vript-Hard dataset or Vriptor model if required.

You must comply with all terms and conditions of these original licenses, including but not limited to the OpenAI Terms of Use, the Copyright Rules & Policies of YouTube or TikTok and the specific licenses for base language models for checkpoints (e.g. Llama-1/2 community license [54, 55], Vicuna [52], and ST-LLM [35]). This project does not impose any additional constraints beyond those stipulated in the original licenses.

# C Vript Dataset Construction

## C.1 Preprocessing

We leverage the PySceneDetect to split the video into scenes by detecting breaks in-between content and moments where the video fades to black. Most of the scenes last from 3s to 1min despite some super long scenes. For each scene, we sample different numbers of frames according to the scene duration: 1) 3 frames for shorter than 6s; 2) 4 frames for shorter than 30s; 3) 5 frames for longer scenes.

 **C.2 Automatic Annotation**

We input multiple images as a video into the GPT-4V. Besides the video frames, we transcribe the voice-over into text using the Whisper model of medium size implemented by FasterWhisper [5]. As shown in Table 4, we use the frames along with the transcription and the video title as the entire input of the video. We also use Claude 3 Sonnet which has a looser constraint on the video content to annotate the remaining scenes that GPT-4V refuses to give a response.

Table 4: An example of the prompt for generating captions in Vript.

```
System: You are an excellent video director that can help me analyze
the given video clip.
```

```
User:  <frame 1> <frame 2> ...  <frame n>
Voice-over:"{voice-over}"
Based on the voice-over and successive frames from the video titled
"{title}" above, please describe:
1) the shot type (15 words)
2) the camera movement (15 words)
3) what is happening as detailed as possible (e.g.  plots,
characters' actions, environment, light, all objects, what they look
like, colors, style, etc.)  (150 words)
4) Summarize the content to title the scene (10 words)
Directly return in the json format like this:  {"shot_type":  "...",
"camera_movement":  "...", "content":  "...", "scene_title":  "..."}.
Do not describe the frames individually but the whole clip.
```

Table 5: Training hyperparameters of Vriptor

| Config | Stage 1 | Stage 2 |
|---|---|---|
| input frame | 16 | 64 |
| input resolution | 224 | 224 |
| max voice-over length | 512 | 2048 |
| max output length | 1024 | 4096 |
| rope scaling factor | 1.0 | 4.0 |
| rope scaling type | - | dynamic |
| learning rate | 2e-5 | 2e-5 |
| learning rate schedule | constant | constant |
| warmup ratio | 0.03 | 0.05 |
| batch size | 128 | 64 |
| epoch | 1 | 1 |
| Qformer state | frozen | frozen |
| Qformer queries | 32 | 32 |
| ViT state | frozen | frozen |

# D   Vriptor Training

Based on the ST-LLM [35], we continue training the model in two stages using the paradigms mentioned in Section 4. At stage 1, for type 3) and type 4) in Figure 2 of multiple scenes, we sample 2~6 successive scenes and concatenate them to form a long video. By doing concatenation, we

---

[5] https://github.com/SYSTRAN/faster-whisper

additionally synthesize 200K long videos and corresponding "sub-scripts", dubbed Vript-Extend. If there are keywords ("voice-over", "say", "narrative", etc) in the captions, we append the voice-over transcription to the end of video frames as the input. We train the model for 1 epoch on Vript and Vript-Extend with a total of 600k video clips, which costs about 500 A100 80GB GPU hours. At stage 2, we continually train the model of stage 1 to empower it to generate dense captions for significantly longer videos. We sample 9~20 successive scenes and synthesize 20K video clips that are much longer than stage 1. As shown in Table 5, we quadruple the input frames to 64. We train on longer videos incorporating 3% of replay data from stage 1 for 1 epoch, which costs about 60 A100 80GB GPU hours.

In Figure 9 and Figure 10, we showcase some examples of the captions generated by Vriptor. Vriptor is capable generate general or dense descriptions for both short (<20s) and long videos (>1min).

Table 6: The full results of various models in Vript-HAL. "voice" means whether the voice-over transcription is utilized for captioning.

| Model | Vript-HAL | | |
| --- | --- | --- | --- |
| | Precision | Recall | F1 |
| BLIP2 [36] | 77.9 | 2.5 | 4.8 |
| InstructBLIP [37] | 71.0 | 13.1 | 21.8 |
| Qwen-VL [38] | 68.8 | 7.2 | 12.4 |
| LLaVA 1.6 34B [1] | 67.6 | 26.3 | 37.8 |
| VideoChatGPT [20] | 69.4 | 9.5 | 16.7 |
| Video-LLaMA [32] | 65.1 | 15.1 | 24.5 |
| VideoChat [31] | 59.1 | 20.9 | 30.9 |
| VideoChat2 [7] | 73.2 | 8.6 | 15.4 |
| ST-LLM [35] | 67.1 | 17.2 | 27.3 |
| PLLaVA 7B [39] | - | - | 32.8 |
| VILA-1.5 8B [40] | - | - | 31.8 |
| Claude 3-Sonnet [41] | 71.7 | 25.1 | 37.2 |
| Claude 3-Opus [41] | 73.4 | 29.1 | 41.7 |
| GPT-4V [15] | 77.3 | 36.2 | 49.3 |
| Claude 3.5-Sonnet [41] | - | - | 44.6 |
| Gemini-1.5-Pro [42] | - | - | 27.0 |
| GPT-4O [43] | - | - | 49.3 |
| GPT-4O (1 fps/100 fr) | - | - | 49.6 |
| Vriptor-W/voice (Ours) | **79.1/80.3** | 26.8/27.7 | 40.0/41.1 |
| Vriptor-S/voice (Ours) | 73.4 /76.1 | **46.1/47.2** | **56.6/58.3** |

# E  Vript-Hard Construction

## E.1  Vript-HAL

**Data Construction**  In order to build Vript-HAL with detailed and high-quality ground truth captions, we carefully select meaningful video clips and annotate the clips with GPT-4V. The meaningful clips here mean that the clips contain several scenes or various events and last longer than 10s, which are filtered by humans. For each clip, we extract ten high-resolution frames, where ten is the maximum number of images allowed for the input of GPT-4V. We input these frames along with a prompt that makes GPT-4 output longer captions containing more details than those in the Vript training dataset. As the GPT-4V sometimes generates captions with hallucinations, to ensure

the reliability of Vript-HAL, we carefully revise the hallucinations and additionally add more details to captions by watching the clip **manually**. We annotate each clip twice using two distinct sampling strategies. The first strategy samples at 5%, 15%, ..., 85%, 95% of the clip and the second samples at 1%, 10%, ..., 80%, 90% of the clip. We make sure that two captions for every clip contain most of the details in the clips so that the calculation of precision score for hallucination evaluation is reliable. If we merge two captions into one, it can be considered as a longer caption of approximately 400 unique words, which would be 40x longer than the captions in MSR-VTT [16] and 20x longer than Panda-70M [14].

**Panda-70M**

The view from the handlebars of a motorcycle as it drives down a street.

**Vript-HAL**

The video clip features a rider operating a motorcycle, presumably an Indian brand bike, from a first-person perspective. The bike has a classic design with chrome detailing and black leather elements. The rider's hands are visible, wearing bright gloves, manipulating the bike's controls. We see a clear windshield, a well-kept dashboard with gauges, and the front part of the bike, including the headlight, which may have extra lights - a detail the voice-over is uncertain about. The environment is a suburban street during the daytime with green lawns, houses, and passing cars. The sky is overcast. The rider comments on starting the bike with the choke and their perplexity about the 'Indian' aspect of the motorcycle, while also noting its coolness. There are no other characters in sight; the focus is on the rider's experience and interaction with the bike.

Figure 6: Comparison between the ground truth captions in Panda-70M and Vript-HAL.

566

### E.2 Vript-RR

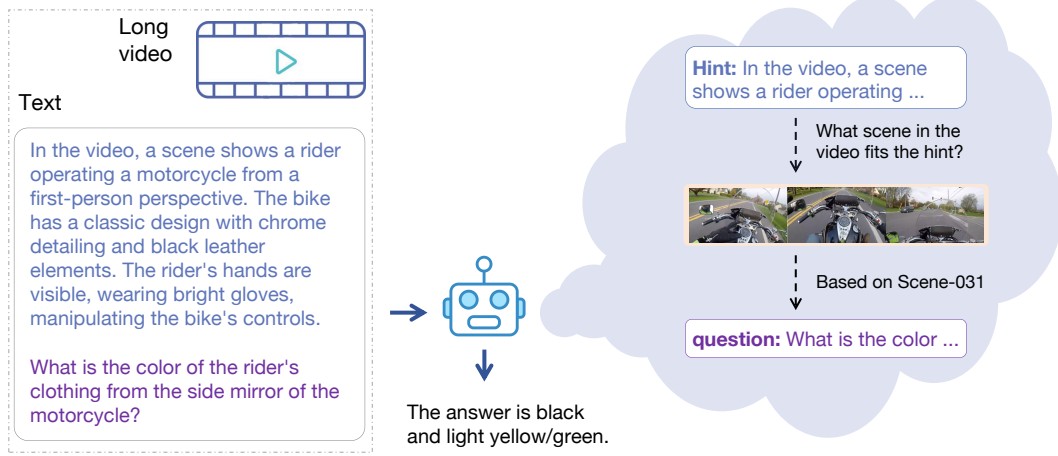

Figure 7: The overview of answering the question in Vript-RR, which is an end-to-end process.

**Data Construction** Each piece of data in Vript-RR consists of a video, a hint, and a corresponding question. The hint and the question are related to a certain scene in the long video. Therefore, we first extract scenes from the video and specially extract four scenes in the 15%, 40%, 60%, and 85% of the video separately to construct four questions at different timestamps per video. We construct four questions per video instead of one question per video because we also want to explore if the temporal positions of the scenes in the video will influence the results of Vript-RR, as illustrated in Section 5.2.

We leverage GPT-4V to generate the hints and the questions for extracted scenes. Given a description of the extracted scene, GPT-4V is prompted to first mask a certain object or character in the description and then ask a question about the masked part. We leverage the masked description generated by GPT-4V as the hint. However, most of the questions generated can not meet the standard of Vript-RR. Humans filter and revise most of the generated questions and hints to make up the Vript-RR finally.

**Data Composition**   As shown in Figure 7, the model accepts the input consisting of a long video, a hint, and a question. The model has to first retrieve the related scene according to the hint and then answer the question. As shown in Table 7, we design various questions that evaluate models' different abilities. Each question requires at least one step of processing or reasoning rather than simply watching the video, which is challenging for most video LLMs.

### E.3   Vript-ERO

**Data Construction**   We sample three unique scenes that only happen once from the long videos (lasting 2min to 2h). Each scene lasts for 10s on average. As shown in Figure 8, we input the descriptions of the shuffled scenes along with the long video and ask the model the give the correct temporal order.

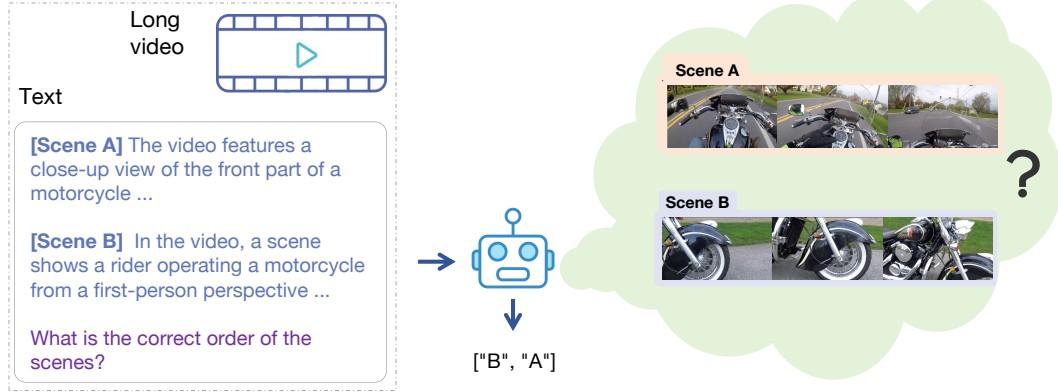

Figure 8: The overview of answering the question in Vript-ERO.

Table 7: Examples of questions in Vript-RR.

| Category | Hint | Question | Answer |
|---|---|---|---|
| Object | . . . a gas station comes into view on the right side of the road with a label "50%" visible at the bottom . . . | What is the name of this gas station visible in the distance? | Shell gas station |
| Position | A man wearing a black shirt with red and white text, is likely affiliated with a brand or eatery . . . | There is an old woman with white hair wearing a black jacket sitting right behind the man, what is she doing? | having a meal |
| Text | . . . spread ideas worth sharing. There's an image being projected which includes a title card featuring a name . . . | What is the name of the speaker of this presentation? | Adam Bernier |
| Color | . . . a rider operating a motorcycle from a first-person perspective. The bike has a classic design. . . | What is the color of the rider's clothing from the side mirror of the motorcycle? | black and light yellow/green and grey |
| Count | A diverse group of individuals, possibly co-workers, are lined up with a row and dressed in casual business attire . . . | Which is more in the scene shown, girls or boys? | girls |
| Implicit | A person is capturing himself and partially other individuals beside him . . . | What object is the left hand of the person holding? | camera |
| Emotion | . . . an individual inside an older model car on the railway, with his hands pressed against the window in a gesture . . . | What emotion does this gesture convey? | urgency or distress or fear |
| Action | . . . park's environment. It's focused on a pair of individuals engaged in a shared activity, sitting on . . . | What are these two individuals doing while sitting on the bench-like structure? | watching the smartphones |
| Fact | . . . a casually dressed man in dark colors is seen loading items into a spacious trunk . . . | What is the brand of the white vehicle? | Chevrolet |
| Cognition | . . . a central figure in black who is receiving touches and hugs . . . | What is the likely scenario or event that this central figure is experiencing? | being eliminated from a show |

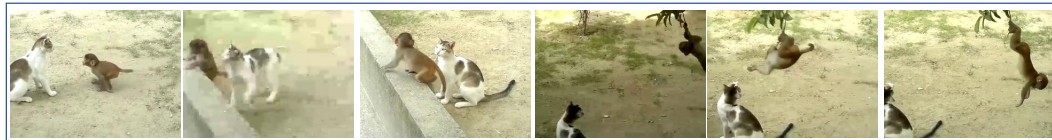

**Vriptor-W (whole-video)**

The video features an outdoor setting with natural lighting, showcasing a playful interaction between a cat and a monkey. The environment appears to be a concrete area with some greenery in the background, suggesting an urban park or garden. The cat is white with brown patches, while the monkey has a reddish-brown fur coat. In the first frame, the cat is standing on its hind legs, front paws reaching up towards the monkey as if trying to climb it. In the second frame, the cat is sitting on the ground, looking up at the monkey with an attentive stance. The third frame captures the cat from behind, gazing towards the monkey, who is now hanging upside down from a tree branch. The final frame shows the monkey swinging towards the cat from the branch.

**Vriptor-S (scene-by-scene)**

**[Scene 1/3: Playful Encounter Between Cat and Monkey](Duration: [0.0, 5.2]/17.0s)**
The video features a cat and a monkey interacting in an outdoor setting, likely a garden or yard with dirt ground and sparse greenery. The cat, with white fur and black patches, is initially seen walking away from the camera towards the monkey. The monkey, with light brown fur, appears to be attempting to climb onto the cat's back. However, the cat seems uninterested and walks past the monkey without acknowledgment. Subsequently, the cat turns around and faces the monkey, who is now standing on its hind legs trying to reach the cat's head. The environment is brightly lit by natural daylight, suggesting it's daytime. There are no other significant objects or characters in the scene.

**[Scene 2/3: Curious Cat Observes Monkey's Climbing Attempt](Duration: [5.2, 12.7]/17.0s)**
The scene takes place outdoors, likely in a garden or yard with dirt ground and sparse grass. A monkey is attempting to climb a concrete wall, while a cat watches from the ground. The monkey, with light brown fur and dark limbs, struggles to ascend the vertical surface. The cat, with white fur and black patches, observes the monkey's actions with interest. In the background, there are trees and bushes indicating a natural environment. The lighting suggests it's daytime with natural sunlight. As the monkey fails to climb the wall, it falls onto the ground. The cat remains stationary throughout the sequence. The colors in the scene are muted with earthy tones dominating due to the natural setting.

**[Scene 3/3: Cat Observes Monkey's Hanging on the Tree](Duration: [12.7, 17.0]/17.0s)**
In an outdoor setting with dirt ground and sparse vegetation, a monkey is attempting to climb a tree branch. The monkey, with light brown fur and dark limbs, struggles to grip the branch. A cat, with white fur and black patches, watches intently from the ground. The cat's attention is focused on the monkey's actions. The environment appears to be a garden or yard during the daytime, with natural sunlight casting shadows on the ground. There are no other objects or characters in the immediate vicinity, drawing the viewer's attention solely to the interaction between the monkey and the cat. The colors in the scene are muted, with earthy tones dominating due to the natural setting.

Figure 9: An example of the caption generated by Vriptor.

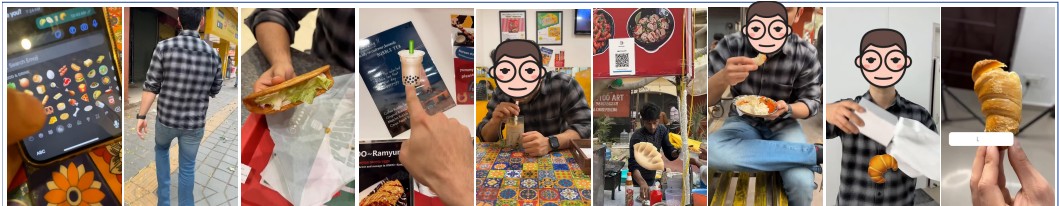

**Vriptor-S (scene-by-scene)**

**[Scene 1/6: Casual Stroll and Burger King Visit](Duration: [0.0, 5.9]/45.9s)**
The clip opens with a close-up of a hand holding a smartphone, displaying an array of colorful emojis on the screen. The focus then shifts to a medium shot of a person from behind as they walk down a street lined with various shops and pedestrians. The person is wearing a plaid shirt and jeans, suggesting a casual outing. The environment appears to be an urban setting during daytime, with clear skies and ample sunlight enhancing the vibrant colors of the surroundings. As the camera follows the person's movement, it captures glimpses of local life and businesses, including what seems to be a Burger King restaurant in the background, indicating a commercial area. The overall atmosphere is lively and bustling with activity.

**[Scene 2/6: Exploring New Delhi's Street Food Scene](Duration: [5.9, 13.0]/45.9s)**
The clip begins with a medium close-up of a person sitting at a table in what appears to be a fast-food restaurant, given the presence of a menu and food packaging. The individual is interacting with someone off-camera, possibly ordering food. The scene transitions to a point-of-view shot where the person is holding a green burger wrapped in paper, about to take a bite. The environment suggests an indoor setting with artificial lighting that casts a warm hue over the scene. The final shot is a close-up of a hand pointing at a sign advertising 'Bubble Tea', indicating the exploration of local street food options. The sign is colorful with red accents, and there's a glimpse of a brightly patterned tablecloth, suggesting a casual dining atmosphere.

**[Scene 3/6: Enjoying Bubble Tea in Cozy Cafe](Duration: [13.0, 17.8]/45.9s)**
The video features a close-up of a person's hand holding a glass filled with a creamy, frothy beverage, likely bubble tea, given the visible tapioca pearls. The drink is served in a clear glass with a straw, placed on a table with a colorful patterned tablecloth. The environment suggests a casual dining setting, possibly a cafe or street food stall. The lighting is bright and natural, indicating daytime. In the background, there are indistinct chatter and ambient sounds that suggest other patrons are present. The person is wearing a plaid shirt, suggesting a laid-back or casual attire. The overall color palette consists of warm tones from the beverage and cooler hues from the surroundings.

**[Scene 4/6: Busy Street Food Stall Ambiance](Duration: [17.8, 23.0]/45.9s)**
The clip opens with a medium shot of a busy street food stall named 'Dona Orginal'. A vendor is seen preparing food, surrounded by various cooking utensils and ingredients. The environment is bustling with activity; people can be seen walking by in the background, indicating a lively urban setting. The lighting is natural, suggesting daytime, and the colors are vibrant, with the reds of the stall contrasting against the more muted tones of the surroundings. The vendor is dressed in casual clothing, focused on his task. The camera then cuts to a close-up of a hand holding a small aluminum foil container filled with dumplings, showcasing the food item in detail. The final shot is a close-up of a person seated outdoors, holding and eating a dumpling, emphasizing the food's texture and taste.

**[Scene 5/6: Tasting Creamy Croissant at Outdoor Eatery](Duration: [23.0, 29.6]/45.9s)**
The scene takes place outdoors, likely in a casual dining area with ambient daylight providing natural illumination. A person is seated, wearing a checkered shirt, and holding a croissant. The croissant appears to be freshly baked, with a golden-brown crust indicative of a flaky pastry. As the person bites into the croissant, the voice-over expresses approval of its creamy interior. The environment seems relaxed with other patrons in the background, suggesting this might be a street food setting or an open-air cafe. The colors are warm and inviting, with the golden hue of the croissant standing out against the more muted tones of the surroundings.

**[Scene 6/6: Leaving a Commendable Croissant](Duration: [29.6, 45.9]/45.9s)**
The scene takes place indoors, likely in a kitchen or dining area, evidenced by the presence of a white refrigerator in the background. The subject, wearing a checkered shirt, is holding a croissant, which they appear to be enjoying. The croissant is golden-brown, indicating it might be freshly baked. As they take bites, the voice-over expresses appreciation for the croissant's quality, suggesting it's commendable. The lighting is bright and natural, suggesting daytime. No other objects or people are in focus, keeping the viewer's attention solely on the croissant and the subject's interaction with it. The colors are warm, with the golden hue of the croissant contrasting against the neutral tones of the room and the subject's checkered shirt.

Figure 10: Another example of the caption generated by Vriptor.

