# OpenReview forum: "Vript: A Video Is Worth Thousands of Words"
_NeurIPS.cc/2024/Datasets_and_Benchmarks_Track — NeurIPS 2024 Track Datasets and Benchmarks Poster_

### Official Review · Reviewer_8smn · 2024-07-13

**Rating:** 8
**Confidence:** 4
**Correctness:** Yes
**Clarity:** Yes

**Review:**

1. This paper is well-written and easy to follow.
2. The probelm to focus on is challenging.
3. The core contributions of this paper are profound and essential.

**Strengths:**

See above

**Additional Feedback:**

N/A

**Documentation:**

Yes

**Limitations:**

Yes

**Opportunities For Improvement:**

1. It's better to provide more detailed statistics about the Vript dataset, such as distributions of topics, distributions of the caption length and etc.
2. It's  better to provide a brief description of the used model in the manuscript.
3. How about the accuracy of PySceneDetect in sepearating scenes in the video? Besides, how to gurantee the scenes could correspond to the Voice-over Transcription?
4. Can Vript be used for training and evaluation?

**Relation To Prior Work:**

Yes

**Summary And Contributions:**

The contribution of this paper is three-fold, including a new dataset of long captions, a new model able to combine many scenes and a hard video benchmark.

---

> ### Author Rebuttal · Authors · 2024-08-18
>
> We are truly grateful for your thoughtful questions and suggestions.
>
> W1: It's better to provide more detailed statistics about the Vript dataset, such as distributions of topics, distributions of the caption length, etc.
>
> A1: We appreciate your attention to the dataset details. **We have indeed provided comprehensive statistics in Figure 1 of our paper, encompassing distributions of topics, caption lengths, durations, and resolutions.** These statistics offer a thorough overview of the Vript dataset's characteristics. We provided more details about Vript in Section 3 and Appendix C. For more meta information about the videos, we provide detailed meta information on [huggingface](https://huggingface.co/datasets/Mutonix/Vript/tree/main/vript_meta) for each video in Vript, including the video ID, title, description, category, duration, resolution, and release date, etc.
>
> W2: It's better to provide a brief description of the used model in the manuscript.
>
> A2: We appreciate this suggestion and agree that it would enhance the paper's clarity. In the revised version, we will incorporate concise yet informative descriptions of the models used in our study in the Appendix. This addition will provide readers with a more comprehensive understanding of our methodological approach.
>
> W3: How about the accuracy of PySceneDetect in separating scenes in the video? Besides, how to guarantee the scenes could correspond to the Voice-over Transcription?
>
> A3: For splitting the video into clips, there is no single and correct solution, which depends on how long each clip lasts and where you want to split (e.g., you may or may not want to split at slow transitions or shot changes). There are several hyperparameters and methods for PySceneDetect to tune the threshold of splitting the videos. In our practice, we use the _AdaptiveDetector_ that calculates frame scores with _ContentDetector_, and then applies a rolling average when processing the result that can help mitigate false detections in situations such as camera movement. We tune the hyperparameter (including `adaptive_threshold`, `min_scene_len`, and `min_content_val`) of _AdaptiveDetector_ to control the duration of each clip to be not shorter than 3s and be not too long.
>
> To guarantee the scenes could correspond to the voice-over, we first split the video into clips (containing both video and audio tracks), and then transcribe the voice-over in the audio track of the clips into text.
>
> W4: Can Vript be used for training and evaluation?
>
> A4: Yes, we use the Vript dataset to train a video captioning model Vriptor in Section 4 in the paper, which **reaches SOTA performance in video captioning**. In practice, **Vript can be used for training strong foundation models of video understanding and video generation**. For example, HPC-AI Tech has used Vript to train video generation model **[OpenSora](https://github.com/hpcaitech/Open-Sora)**. For evaluation, we recommend using the Vript-hard benchmark introduced in Section 5 to evaluate the model hallucinations, reasoning and long video understanding ability.
>
> We sincerely thank the reviewer for their positive feedback and insightful questions. We are pleased that our paper was well-received, and we appreciate the opportunity to clarify and expand on certain aspects of our work.

---

> > ### Comment · Reviewer_8smn · 2024-08-19
> > **Response**
> >
> > Thanks for the rebuttal. I will keep my initial score.

---

### Official Review · Reviewer_CXVg · 2024-07-24
**Review of Vript**

**Rating:** 5
**Confidence:** 4
**Correctness:** Yes.
**Clarity:** Yes.

**Review:**

This work presents a high-quality video-text dataset and introduces innovative benchmarks, which are significant for advancing video understanding research. However, the paper's organization requires polishing. It omits crucial details about the framework and training pipeline and contains ambiguous concepts, such as the `three training paradigms`. The presentation of results is incomplete. Particularly in Table 3, only partial results of `Vript-ERO` are shown, and inconsistencies in QA ranks are noted. Despite these weaknesses, the originality and potential impact of the dataset and benchmarks are commendable.

**Strengths:**

- Comprehensive Dataset: Vript offers a richly annotated dataset with 420K clips, each with dense, script-like captions that include camera operations, enhancing the depth and quality of video descriptions.
- Benchmark Introduction: The introduction of Vript-Hard benchmarks addresses gaps in existing video understanding tasks by focusing on hallucination evaluation, long-video QA, and event re-ordering, which are more challenging and detailed.
- State-of-the-art Performance: Vriptor demonstrates superior performance in video captioning tasks, comparable to advanced models like GPT-4V, indicating the effectiveness of the dataset and training paradigms.

**Additional Feedback:**

See limitations.

**Documentation:**

Yes.

**Ethics:**

No.

**Limitations:**

Yes.

**Opportunities For Improvement:**

- Paper Organization:
  - Besides the dataset, the authors also fine-tune a model. However, the details of the entire framework and training pipeline are not provided in the main text.
  - Moreover, some concepts are ambiguous. For instance, `three training paradigms`, does it mean three types of SFT data?
  - The `Precision`, `Recall`, `F1` should be stated before Table 2.
- Missing results:
  - In Table 3, the authors only provide part of the results of `Vript-ERO`. Also, the frame numbers are not clear, which is essential for long videos.
  - In Table 3, the ranks of open-ended and multiple-choice QAs are not consistent. Why?
- Missing datasets in Table1:
  - YT-Temporal (Merlot: Multimodal neural script knowledge models)
  - InternVid: A Large-scale Video-Text Dataset for Multimodal Understanding and Generation
  - FAVDBench: Fine-grained Audible Video Description

**Relation To Prior Work:**

Yes.

**Summary And Contributions:**

The paper presents Vript, a new video-text dataset featuring 12K high-resolution videos with detailed captions averaging 145 words per clip. It introduces three challenging benchmarks: Vript-HAL (Hallucination Evaluation), Vript-RR (Retrieval and Reasoning), and Vript-ERO (Event Re-ordering). The dataset and its accompanying model, Vriptor, aim to improve video captioning and video understanding, achieving state-of-the-art performance among open-source models.

---

> ### Author Rebuttal · Authors · 2024-08-18
>
> We appreciate the time and effort you’ve put into formulating such insightful questions and suggestions.
>
> ### Paper Organization
>
> W1: The authors should also fine-tune a model. However, the details of the entire framework and training pipeline are not provided in the main text.
>
> A1: The main focus of this work is **the foundation model of video understanding by exploring better vision-text alignment**. Based on the better vision-text alignment, we plan to finetune the model for comprehensive ability or explore further applications in video generation, e.g, several research groups like HPC-AI Tech have used Vript to train their video generation models like **[OpenSora](https://github.com/hpcaitech/Open-Sora)**.
>
> Due to space limitations, **we provide the details of the entire framework and training pipeline in Appendix D**. For more details, we provide detailed instructions and code for reproducing the entire training of our models in [here](https://github.com/mutonix/Vript/tree/main/vriptor).
>
> We appreciate the reviewer's suggestion and will consider including more details in the main text in the revised version for better clarity.
>
> W2: Three training paradigms, does it mean three types of SFT data?
>
> A2: Yes, the three training paradigms correspond to three types of SFT data. Each type of SFT data represents a novel approach in video-caption formatting, addressing aspects that have been less explored in previous works: (1) Longer videos should correspond to longer captions instead of fixed length, helping the model to output more detailed captions. (2) Voice-over provides extra information to help the model understand what the objects are in the video. (3) Captions with timestamps help the model to know the start and end of each scene, which is important for the model to generate more detailed captions.
>
> W3: The Precision, Recall, and F1 should be stated before Table 2.
>
> A3: We appreciate the reviewer's suggestion. Our original intention in placing the metrics explanation after Table 2 is because we want to introduce them along with Vript-HAL in Section 5 for better understanding. We will revise the paper to include a brief introduction of these metrics before Table 2 for better clarity.
>
> ### Missing results
>
> W4: In Table 3, the authors provide part of the results of Vript-ERO. Also, the frame numbers are not clear.
>
> A4: **As we have illustrated in L269 in the paper, in Table 3, we do not provide part of the results because these models fail to give answers in Vript-ERO**. Different from previous tasks that only have short questions, Vript-ERO also contains long descriptions of scenes. **Many models fail in Vript-ERO because they are weak in processing long instructions and only output repetitive answers**.
>
> For the number of frames in Vript-ERO, we test the open-source models using 16 frames (following the VideoChat2 setting) and the closed-source models using 10 frames (because GPT-4V only accepts maximum 10 frames). For more details, we provide detailed instructions and examples of how to evaluate the models on the Vript-hard in [here](https://github.com/mutonix/Vript/tree/main/vript-hard).
>
>
> W5: In Table 3, the ranks of open-ended and multiple-choice QAs are not consistent. Why?
>
> A5: To be specific, **only VideoChat2 has inconsistency in the ranks** of open-ended and multiple-choice QAs with higher accuracy in multiple-choice QAs. Because MVBench (proposed in the VideoChat2 paper) consists of multiple-choice QAs, we hypothesize that VideoChat2 has been particularly reinforced in multiple-choice questions. It is reasonable that the model has better performance in multiple-choice QAs than in open-ended QAs if it is trained more on multiple-choice QAs.
>
> ### Missing datasets in Table 1
>
> We supplement these datasets and will include them in the revised version of the paper.
>
> | |Domain | Text Len | Clips | Duration | Resolution | Method |
> | --- | --- | --- | --- | --- | --- | --- |
> | YT-Temporal-180 [1] | Open | ~10 | 180M | - | 480p | ASR |
> | InternVid [2] | Open | 17.6 | 234M | 760.3Kh | 720p | BLIP |
> | FAVDBench [3] | Open | 218.9 | 11.4K | 24.4h | 720p-2K | AVLFormer [3] |
> | Vript (ours) | Open | 145 | 420K | 1.3Kh | 720p-2K | GPT-4V |
>
> [1] Zellers, Rowan, et al. "Merlot: Multimodal neural script knowledge models."
>
> [2] Wang, Yi, et al. "Internvid: A large-scale video-text dataset for multimodal understanding and generation."
>
> [3] Shen, Xuyang, et al. "Fine-grained audible video description."
>
> We sincerely appreciate the reviewer's thorough evaluation and insightful comments. We believe that addressing these points will significantly improve the clarity and completeness of our paper. We are committed to incorporating these suggestions in the revised version, which we believe will strengthen our contribution to the field of video understanding and caption generation.

---

> ### Author Response · Authors · 2024-08-27
> **Could you please check our response to see if we address your concerns?**
>
> Thank you for your thoughtful and detailed feedback. We appreciate the time you've taken to review our work. As the rebuttal period comes to a close, we would be grateful if you could check our response to see if we address your concerns. Your continued guidance is much appreciated.
>
> Best,
>
> Authors

---

### Official Review · Reviewer_n9oD · 2024-07-24
**Valuable contributions with new dataset and benchmarks, SOTA results for image captioning, but missing data details and clarity needed for comprehensive evaluation**

**Rating:** 9
**Confidence:** 4
**Clarity:** The paper is clearly written and well…

**Review:**

#### 4.2 Experiment and Analysis

**Voice-overs Help Model Understand What They Are**
- "We also observe a 14% increment in the proportion of proper nouns of all nouns in the captions." Where is this information presented? What were the numbers before the use of voice-over? This information is not found in Appendix D, as referenced. Please include these numbers in the main paper for easier reference.

**Timestamps Help Model Know the Starts and the Ends**
- "Besides, the model with timestamps gives more detailed captions with a 12% higher recall on Vript-HAL while the model without timestamps is more likely to forget to describe some parts of the videos." Where are these numbers presented? Again, the information is not readily available in Appendix D. Please include these figures in the main paper.

#### 5 Vript-Hard

**E.1 Vript-HAL**

**Data Construction**
- **General**: How many instances are there in Vript-HAL? Please add this information to the main paper.
- "As GPT-4V sometimes generates captions with hallucinations, to ensure the reliability of Vript-HAL, we carefully revise the hallucinations and additionally add more details to captions by watching the clip manually." What are the statistics?
- "We make sure that two captions for every clip contain most of the details in the clips so that the calculation of precision score for hallucination evaluation is reliable." How is this ensured? Please provide more details.

**E.2 Vript-RR**

**General**: How many instances are there in Vript-RR? Please add this information to the main paper.

**Data Construction**
- "However, most of the questions generated cannot meet the standard of Vript-RR. Humans filter and revise most of the generated questions and hints to make up Vript-RR finally." Who were these humans? How many instances have you filtered? Please provide more details.

**E.3 Vript-ERO**

**General**: How many instances are there in Vript-ERO? Please add this information to the main paper.

**Strengths:**

The paper introduces the Vript dataset with detailed captions and the innovative Vriptor video captioning model, alongside the comprehensive Vript-Hard benchmark. The well-documented dataset and quality research, make this work highly relevant.

**Additional Feedback:**

Great work, important contribution.

**Correctness:**

The claims in the submission are correct and the work is generally sound. The dataset and benchmark evaluation methods are appropriate, though providing explicit details on the dataset size and collection process would enhance clarity. Overall, the work is strong and well-executed.

**Documentation:**

Yes, the authors discussed data collection, model training, and sources in both the paper and the appendix. They also provided a URL to a detailed GitHub page: https://github.com/mutonix/Vript.

**Ethics:**

The authors discuss the ethics in the appendix section, including data usage licensing and privacy issues that come from the nature of video collection (social media, etc).

**Limitations:**

The authors have addressed the limitations and potential negative societal impacts of their work.

**Opportunities For Improvement:**

The dataset and benchmarks lack details on instance counts and data collection methodologies (see review). Key statistical information and clarifications on manual revisions and human annotation should be included in the main paper for better transparency and reliability.

**Relation To Prior Work:**

Yes.

**Summary And Contributions:**

The authors construct a video-text dataset called Vript, featuring dense and detailed captions. They train a video captioning model named Vriptor based on this dataset. Additionally, they introduce a video understanding benchmark named Vript-Hard, which consists of three tasks: Vript-HAL, Vript-RR, and Vript-ERO. The paper is well-written and organized, with rich and well-documented data.

---

> ### Author Rebuttal · Authors · 2024-08-18
>
> We sincerely thank you for these insightful questions and suggestions.
>
> ## 4.2 Experiment and Analysis
>
> ### Voice-overs Help Model Understand What They Are
>
> W1: We also observe a 14% increment in the proportion of proper nouns of all nouns in the captions." Where is this information presented? What were the numbers before the use of voice-over?
>
> A1: We use the SpaCy library to detect the proper nouns and nouns in the captions after lemmatization and removing the stop words. Before using the voice-over, the proportion of proper nouns of all nouns is about 2.1% on average. After using the voice-over as the extra information, the proportion increases to 16.2% by 14.1%, which indicates that the model with voice-over data is better at recognizing and generating proper nouns, enhancing the specificity of the captions.
>
> ### Timestamps Help Model Know the Starts and the Ends
>
> W2: "Besides, the model with timestamps gives more detailed captions with a 12% higher recall on Vript-HAL while the model without timestamps is more likely to forget to describe some parts of the videos." Where are these numbers presented?
>
> A2: Due to space limitations, we only provide the conclusion in the main text and experimental details will be supplemented in the Appendix as you advise. To further illustrate, we conduct experiments using Vriptor-S (without voice-over) on Vript-HAL. The Vriptor-S (w/o voice-over) with timestamps has a recall score of 46.1. If we remove the timestamps, the recall score of Vript-S (w/o voice-over) decreases to 41.2. It means that the model without timestamps is more likely to forget to describe some parts of the videos and give less detailed captions.
>
> ## 5 Vript-Hard
>
> ## E.1 Vript-HAL
>
> ### General
>
> W3: How many instances are there in Vript-HAL? Please add this information to the main paper.
>
> A3: There are 122 instances in Vript-HAL and each one is carefully checked by humans, which can be explored on [huggingface](https://huggingface.co/datasets/Mutonix/Vript-HAL).
>
> ### Data Construction
>
> W4: As GPT-4V sometimes generates captions with hallucinations, to ensure the reliability of Vript-HAL, we carefully revise the hallucinations and additionally add more details to captions by watching the clip manually." What are the statistics?
>
> A4: Although most of the instances in Vript-hard are generated by GPT-4V, every instance is double-checked and revised by humans. For example, in Vript-HAL, we require the humans (composed of the authors) to carefully check if there are hallucinations in the captions and add more details based on their observations. We do not collect specific statistics, e.g., how many hallucinations are observed, during the process of re-checking the GPT-4V annotated data. In empirical practice, GPT-4V sometimes generates hallucinations when describing the details in the video, e.g., the man is squatting instead of sitting, so we humans need to revise such hallucinations to make sure the ground truth answers are correct.
>
> W5: "We make sure that two captions for every clip contain most of the details in the clips so that the calculation of precision score for hallucination evaluation is reliable." How is this ensured? Please provide more details.
>
> A5: If we want the precision score for hallucination evaluation to be reliable, we need to ensure that the ground truth answers can include most of the details in the clips. **Therefore, we annotate each clip by annotating two captions twice using GPT-4V plus human checking.**
>
> In Appendix E.1, because GPT-4V only accepts maximum 10 frames, we annotate each clip twice using two distinct sampling strategies to include more frames. The first strategy samples at 5%, 15%, ..., 85%, 95% of the clip (10 frames) and the second samples at 1%, 10%, ..., 80%, 90% of the clip (10 frames). In practice, two captions with different sampling strategies **focus on different details because the sampled frames are different**. Besides that, we carefully revise the hallucinations and additionally add more details to captions by watching the clips **manually**.
>
>
> ## E.2 Vript-RR
>
> ### General
>
> W6: How many instances are there in Vript-RR? Please add this information to the main paper.
>
> A6: There are 152 instances in Vript-RR and each one is carefully checked by humans, which can be explored on [huggingface](https://huggingface.co/datasets/Mutonix/Vript-RR).
>
> ### Data Construction
>
> W7: "However, most of the questions generated cannot meet the standard of Vript-RR. Humans filter and revise most of the generated questions and hints to make up Vript-RR finally." Who were these humans? How many instances have you filtered? Please provide more details.
>
> A7: The human annotators are the authors. We use GPT-4V to generate 560 questions of Vript-RR for 560 video clips. To ensure the diversity of the questions, we filter out 408 similar questions using GPT-4. The humans revise 97 of the remaining 152 questions in Vript-RR to meet the standard (where the questions require multi-step reasoning and ground truth answers should be correct).
>
>
> ### E.3 Vript-ERO
>
> ### General
>
> W8: How many instances are there in Vript-ERO? Please add this information to the main paper.
>
> A8: There are 134 instances in Vript-ERO and each one is carefully checked by humans, which can be explored on [huggingface](https://huggingface.co/datasets/Mutonix/Vript-ERO).
>
> In conclusion, we will incorporate all these details into the revised version of our paper to enhance clarity and transparency.

---

> > ### Comment · Reviewer_n9oD · 2024-08-29
> >
> > Thanks for the rebuttal, I will maintain my current score.

---

### Official Review · Reviewer_65SF · 2024-07-27

**Rating:** 6
**Confidence:** 4
**Correctness:** Yes
**Clarity:** Yes

**Review:**

The paper is mostly well-written and easy to follow. The proposed dataset and benchmark address a timely topic in LVMs research and are good contributions. Some improvements could be made in describing how the dataset is constructed and two crucial representative baselines are missing in Vript-Hard.

Please see Strengths and Opportunities For Improvement for pros and cons of this work. I am willing to increase my scores if the cons are properly addressed.

**Strengths:**

- The Vript dataset has long, detailed captions and is a valuable contribution to the research for LVMs.
- The Vriptor model achieves competitive results on video captioning tasks.
- The proposed Vript-Hard benchmark is timely – the evaluation for hallucination, long video retrieval & understanding, and temporal understanding is important but mostly lacking in previous benchmarks. The performance gap between open-source models and proprietary models indicates the value of this benchmark.

**Additional Feedback:**

N/A

**Documentation:**

Mostly sufficient. Need more information on data collection and organization.

**Limitations:**

Yes

**Opportunities For Improvement:**

- The paper does not provide enough details on the data collection and cleaning process. It only mentions that the videos are collected from YouTube and TikTok, but no detailed description such as more specified sources, genres, and categories and so on. It is unclear what those videos are and what criteria was used in selecting and filtering them. Please provide more information on this
- Gemini-1.5-Pro and GPT4-O are state-of-the-art LVMs at the time of submission as well as at present. It would be interesting to see how they perform on the challenging Vript-Hard benchmark, which is currently missing in the paper.
- Missing some representative open-source baselines, such as PLLaVA [1] and VILA-1.5 [2]. Should include them in the experiments.
- The number of frames used for each model evaluated is not specified. This is especially crucial for understanding the Vript-RR and Vript-ERO results. Please specify the number of frames used, the context window size of the models, and it would be interesting to see how the model performances will change with different number of input frames.

[1] PLLaVA : Parameter-free LLaVA Extension from Images to Videos for Video Dense Captioning. https://github.com/magic-research/PLLaVA
[2] VILA: On Pre-training for Visual Language Models. CVPR 2024. https://github.com/NVlabs/VILA

**Relation To Prior Work:**

Yes

**Summary And Contributions:**

This paper introduces Vript, a novel video-text dataset comprising 12K high-resolution videos with detailed script-like captions. The authors also present Vriptor, a video captioning model trained on Vript, and Vript-Hard, a challenging benchmark for evaluating video understanding models with hallucination, long video understanding, and temporal understanding.

---

> ### Author Rebuttal · Authors · 2024-08-18
>
> We greatly appreciate your insightful questions and valuable suggestions.
>
> W1: The paper does not provide enough details on the data collection and cleaning process. No detailed description such as more specified sources, genres, and categories and so on is provided. It is unclear what those videos are and what criteria were used in selecting and filtering them.
>
> A1: We appreciate your attention to the dataset details. **In Figure 1 in the paper, we have provided the description of Vript including resolutions, categories, caption length distribution, and durations.** For example, we provide the distributions across 14 different topics, including News, Sports, Blogs, Travel, etc. In L103 of the paper, we introduce the sources of videos are YouTube and TikTok. To be specific, the long videos are selected from YouTube videos in the HD-VILA-100M [1]. The short videos are scraped from the most popular videos on YouTube Shorts and TikTok to avoid including personal information.
>
> For more meta information about the videos, we provide detailed meta information on [huggingface](https://huggingface.co/datasets/Mutonix/Vript/tree/main/vript_meta) for each video in Vript, including the video ID, title, description, category, duration, resolution, and release date, etc.
>
> **For the criteria of selecting and filtering, we select videos based on their categories and release dates to enhance diversity**. From the pie of video category in Figure 1, we can see that the video quantities are nearly uniformly distributed across each category, which indicates the consideration of diversity during the selection. But we also have preferences in selection, for example, we reduce the number of music videos and increase the number of videos of blogs and travels to include more people and landscapes.
>
> We do not filter the videos based on the content or duration for diversity, e.g., we include both short (<10s) and long videos (>2h) in Vript. We filter the video that contains NSFW content or is not suitable for general audiences.
>
> In conclusion, our data collection process was carefully designed to ensure diversity and quality across various dimensions including video sources, categories, etc.
>
> [1] Xue, Hongwei, et al. "Advancing high-resolution video-language representation with large-scale video transcriptions."
>
> W2: Closed-source models like Gemini-1.5-Pro and GPT4-O performance on Vript-hard.
>
> A2: **We have supplemented the Vript-hard benchmark with results from closed-source models including Gemini-1.5-Pro, GPT4-O, and Claude-Sonnet-3.5, as shown in Table 1 below**. We are taking more closed-source models into account to be included in the Vript-hard benchmark.
>
> Table 1. Gemini-1.5-Pro, GPT4-O, Claude-Sonnet-3.5 performance on Vript-hard.
> |     | HAL | RR (Scene-M) | RR (Scene-O) | RR (Whole-M) | RR (Whole-O) | ERO (@1) | ERO (@2)|ERO (@3) |
> | --- | --- | --- | --- | --- | --- | --- | --- | --- |
> | GPT4-O | 49.27 | 92.11 | 77.48 | 72.37 | 54.61 | 75.0| 32.58 |32.58 |
> | Gemini-1.5-Pro | 27.01 | 85.12 | 83.55 | 59.87 | 57.52 | 35.06| 18.18 | 9.09 |
> | Claude-Sonnet-3.5 | 44.55 | 80.92 | 59.21 | 54.61 | 42.76 | 56.72 | 18.66 | 6.72 |
>
> W3: More open-source models like PLLaVa and VILA-1.5 on Vript-hard.
>
> A3: **We have included results from open-source models such as PLLaVA and VILA-1.5 in the Vript-hard benchmark, as shown in Table 2 below**. We will include more open-source models including InternVL-Chat and LLaVA-Video to our Vript-hard later to make the benchmark more comprehensive.
>
> These results further demonstrate that while closed-source models generally outperform open-source alternatives, there is still significant room for performance improvement in certain tasks, highlighting the challenging nature of our benchmark.
>
> Table 2. PLLaVA and VILA-1.5 performance on Vript-hard. _Both PLLaVA and VILA-1.5 fail in ERO because they are weak in processing long instructions and only output repetitive answers._
> |     | HAL | RR (Scene-M) | RR (Scene-O) | RR (Whole-M) | RR (Whole-O) | ERO (@1/@2/@3) |
> | --- | --- | --- | --- | --- | --- | --- |
> | PLLaVA [2] | 32.82 | 62.50 | 46.05 | 55.26 | 36.18 | fail |
> | VILA-1.5 [3] | 31.75 | 75.00 | 48.68 | 55.26 | 32.24 | fail |
>
>
> [2] Xu, Lin, et al. "Pllava: Parameter-free llava extension from images to videos for video dense captioning."
>
> [3] Lin, Ji, et al. "Vila: On pre-training for visual language models."
>
> W4: Please specify the number of frames used, the context window size of the models, and it would be interesting to see how the model performances will change with different numbers of input frames.
>
> A4: For the number of frames used, we use 10 frames for closed-source models (because GPT-4V only accepts maximum 10 frames) and we use 16 frames for open-source models (following the VideoChat2 settings). We also provide detailed instructions and examples of how to evaluate these models on the Vript-hard in [here](https://github.com/mutonix/Vript/tree/main/vript-hard).
>
> Regarding the context window size, we interpret this as referring to the input lengths of different models. These input lengths vary depending on how each model encodes video frames. For example, ST-LLM uses 32 tokens for one frame, while VideoChat2 uses 64 tokens per frame. This variation in encoding affects the total number of frames that can be processed within a model's maximum context length.
>
> The influence of the number of input frames depends on the model architecture and the specific task. It is interesting to explore the influence of the number of input frames on the model performance. We will leave that for future work.
>
>
>
> We appreciate the reviewer's thoughtful comments. In response, we have provided (1) details on the data collection and cleaning process, (2) more closed-source model results, (3) more open-source model results, and (4) evaluation details including the number of frames. We believe these additions significantly strengthen our paper and address the reviewer's concerns.

---

> ### Author Response · Authors · 2024-08-27
> **Could you please check our response to see if we address your concerns?**
>
> Thank you for your thoughtful and detailed feedback. We appreciate the time you've taken to review our work. As the rebuttal period comes to a close, we would be grateful if you could check our response to see if we address your concerns. Your continued guidance is much appreciated.
>
> Best,
>
> Authors

---

> ### Comment · Reviewer_65SF · 2024-08-28
> **Response to rebuttal**
>
> Thank you for the comprehensive rebuttal. I have some follow-up questions on the experiments:
> 1. Were Gemini-1.5-Pro and GPT4-O also evaluated with 10 frames? They have a much larger context window and if so, I would recommend the authors to evaluate them at 1/0.5 fps or 384 frames as used in [Video-MME](https://video-mme.github.io/home_page.html#leaderboard). I am also wondering if authors could provide an explanation on why Gemini-1.5-Pro's result is so much worse than GPT4-O, since this was usually not the case.
> 2. Could you specify which PLLaVA and VILA-1.5 models were used? Especially the number of parameters.

---

> > ### Author Response · Authors · 2024-08-29
> > **Response to Reviewer 65SF’s comment on rebuttal**
> >
> > 1.Were Gemini-1.5-Pro and GPT-4O also evaluated with 10 frames? They have a much larger context window and if so, I would recommend the authors to evaluate them at 1/0.5 fps or 384 frames as used in Video-MME.
> >
> > A: Yes, all closed-sourced models are evaluated with 10 frames for fair comparison (because GPT-4V can only accept maximum 10 frames). In light of the discussion on the larger context windows for models, we evaluate Gemini-1.5-Pro and GPT-4O with more frames on Vript-Hard. Specifically, if the video is shorter than 100s, we sample the video at 1 fps. If the video is longer than 100s, we sample 100 frames from the video. It is a pity that currently we do not have enough budget for Gemini-1.5-Pro to run all the tasks in Vript-Hard, especially for many long videos in RR and ERO. Therefore, we only evaluate Gemini-1.5-Pro with 1 fps/100 frames on HAL. We will establish a separate leaderboard for these long-context models by evaluating them with more frames in the future.
> >
> > Table 1. Gemini-1.5-Pro, GPT-4O on Vript-hard with more frames. We will fill in the blank "-" and update more models in the future.
> > |     | HAL | RR (Scene-M) | RR (Scene-O) | RR (Whole-M) | RR (Whole-O) | ERO (@1) | ERO (@2)|ERO (@3) |
> > | --- | --- | --- | --- | --- | --- | --- | --- | --- |
> > | GPT-4O (10 frames) | 49.27 | 92.11 | 77.48 | 72.37 | 54.61 | 75.0| 32.58 |32.58 |
> > | GPT-4O (1 fps/100 frames) | 49.62 | 91.36 | 78.23 | 78.51 | 65.98 | 81.06 | 40.15 | 38.63 |
> > | Gemini-1.5-Pro (10 frames) | 27.01 | 85.12 | 83.55 | 59.87 | 57.52 | 35.06| 18.18 | 9.09 |
> > | Gemini-1.5-Pro (1 fps/100 frames)  | 27.26 | - | - | - | - | - | - | - |
> >
> > Comparing the performance of GPT-4O with 10 frames and 1 fps/100 frames, we observe that the performance of GPT-4O increases significantly higher on RR (whole video) than RR (scene). For event re-ordering in long videos (ERO), the performance of GPT-4O increases with more frames. More frames provide more context for the model to get better performance on long videos.
> >
> >
> > 2.I am also wondering if authors could provide an explanation on why Gemini-1.5-Pro's result is so much worse than GPT-4O, since this was usually not the case.
> >
> > A: While Gemini-1.5-Pro exhibits superior performance compared to GPT-4O in [Video-MME](https://video-mme.github.io/home_page.html#leaderboard), **the tasks in Vript-Hard differ significantly from those in Video-MME, leading to distinct outcomes**. Specifically, for the Vript-HAL evaluating hallucinations, Gemini-1.5-Pro fails to generate captions with the same level of detail as GPT-4O with 10 frames and also fails with 1 fps/100 frames. For the Vript-RR, which involves multi-step reasoning, Gemini-1.5-Pro excels at providing more detailed CoT explanations for open-ended questions but performs less well on close-ended questions, possibly due to its greater reliance on CoT. Regarding Vript-ERO, Gemini-1.5-Pro occasionally declines to answer certain questions by stating, "Please provide the video", even when a video is indeed supplied. In several cases we have tested, this phenomenon is also observed using 1 fps/100 frames. This behavior might be attributed to the inclusion of questions featuring lengthy scene descriptions, which diverge from the typical Vision-Language QA format where questions are generally brief.
> >
> >
> > 3.Could you specify which PLLaVA and VILA-1.5 models were used? Especially the number of parameters.
> >
> > A: We use [PLLaVA 7B](https://huggingface.co/ermu2001/pllava-7b) and [VILA-1.5 8B](https://huggingface.co/Efficient-Large-Model/Llama-3-VILA1.5-8B) with similar sizes for comparison.

---

> ### Comment · Reviewer_65SF · 2024-09-01
> **Response to comment**
>
> Thank you for the follow-up comments. I would increase my score to 6.

---

### Decision · Program_Chairs · 2024-09-26

**Decision:**

Accept (Poster)

**Comment:**

The paper introduces Vript, a novel video-text dataset comprising 12K high-resolution videos with detailed captions averaging 145 words per clip. It also presents Vriptor, a video captioning model trained on Vript, and Vript-Hard, a challenging benchmark for evaluating video understanding models on tasks like hallucination evaluation, long-video question answering, and event re-ordering.

*Strengths*
+ Valuable dataset contribution: Vript offers long, detailed captions, a significant resource for research in large video models (R-65SF, R-CXVg, R-8smn)
+ Competitive model performance: Vriptor achieves SOTA results in video captioning, demonstrating effectiveness among open-source models and comparability to advanced models like GPT-4V (R-65SF, R-CXVg, R-8smn)
+ Innovative benchmark: Vript-Hard addresses important aspects of video understanding lacking in previous benchmarks, focusing on hallucination evaluation, long video retrieval, and temporal understanding (R-65SF, R-n9oD, R-CXVg)
+ Clarity and organization: The paper is generally well-written and accessible to readers (R-65SF, R-n9oD, R-8smn)

*Weaknesses and rebuttals*
- Insufficient data collection details: Initially, the paper lacked details on data collection and cleaning processes. The authors have now provided comprehensive information on video sources, categories, selection criteria, and meta-information available on Hugging Face (addressing R-65SF, R-n9oD, R-8smn)
- Missing baselines in experiments: Crucial baselines like Gemini-1.5-Pro, GPT4-O, PLLaVA, and VILA-1.5 were initially absent. The authors have included these in the Vript-Hard benchmark, strengthening experimental comparisons (addressing R-65SF)
- Unclear evaluation metrics and results presentation: The number of frames used and context window sizes were unspecified. The authors clarified these details and provided evaluation instructions and examples (addressing R-65SF, R-CXVg)
- Paper organization and ambiguity: Issues regarding ambiguous concepts and missing details about the framework and training pipeline were raised. The authors clarified these points and agreed to include more details in the main text (addressing R-CXVg)
- Missing Statistical Information: Key statistical information and clarifications on manual revisions and human annotations were initially lacking. The authors have provided these details and will include them in the revised paper. (addressing R-n9oD)

The reviewers recognize the contributions of this paper in introducing a valuable dataset, a SOTA model, and an innovative benchmark in video understanding. As the authors have satisfactorily addressed all concerns, the AC recommends acceptance.